# Evaluation of high-resolution melt curve analysis for rapid differentiation of *Campylobacter hepaticus* from other species in birds

**Petrina Young**[1], **Pol Tarce**[1], **Sadhana Adhikary**[1], **Joanne Connolly**[1], **Tim Crawshaw**[1,2], **Seyed A. Ghorashi**[1]*

1 School of Animal and Veterinary Sciences, Charles Sturt University, Wagga Wagga, New South Wales, Australia, 2 School of Veterinary Sciences, Massey University, Tennent Drive, Palmerston North, New Zealand

* aghorashi@csu.edu.au

## Abstract

Spotty liver disease (SLD) is a bacterial disease of chicken, causing mortalities and reduction in egg production, hence, contributing to economic loss in the poultry industry. The causative agent of SLD has only recently been identified as a novel *Campylobacter* species, *Campylobacter hepaticus*. Specific primers were designed from the *hsp60* gene of *Campylobacter hepaticus* and PCR followed by high-resolution melt curve analysis was optimised to detect and differentiate three species of *Campylobacter* (*Campylobacter coli*, *Campylobacter jejuni* and *Campylobacter hepaticus*). The three *Campylobacter* species produced a distinct curve profile and was differentiated using HRM curve analysis. The potential of the PCR-HRM curve analysis was shown in the genotyping of 37 *Campylobacter* isolates from clinical specimens from poultry farms. PCR-HRM curve analysis of DNA extracts from bile samples or cultures from bile samples, were identified as *Campylobacter hepaticus* and confirmed by DNA sequencing. The DNA sequence analysis of selected samples from each of the three HRM distinctive curves patterns showed that each DNA sequence was associated with a unique melt profile. The potential of the PCR-HRM curve analysis in genotyping of *Campylobacter* species was also evaluated using faecal specimens from 100 wild birds. The results presented in this study indicate that PCR followed by HRM curve analysis provides a rapid and robust technique for genotyping of *Campylobacter* species using either bacterial cultures or clinical specimens.

## Introduction

Spotty liver disease (SLD) is a disease of poultry that causes mortality and reduced egg production. It mainly affects free-range laying hens. On post mortem examination there are multiple 1-2mm pale focal lesions in the liver [1]. Spotty liver disease was first reported from the Eastern states of Australia in the 1980s [2] and subsequently has been reported from Europe,

**Funding:** This work was supported by the Discovery Translation Fund (grant no. DTF327-103074).

**Competing interests:** The authors have declared that no competing interests exist.

North America and Africa [3–7]. Spotty liver disease is similar to avian vibrionic hepatitis (AVH) a disease which was reported in the USA and Canada between the 1950s and 1970s. The causative agent of AVH was never fully defined. Both SLD and AVH have been used to describe an acute, randomly distributed, focal necrotic hepatitis [8].

*Campylobacter hepaticus* (*C. hepaticus*) has recently been identified [9], named [10] and shown to be the cause of SLD [11]. It is fastidious, difficult to isolate and will not grow on many of the selective media used to isolate *Campylobacter* [9].

Spotty liver disease is recognised as a significant threat to the Australian layer industry [12]. Mortality in an affected flock can be 10% with a 10% fall in egg production [1]. The incidence of the disease appears to be increasing and this coincides with changes in the husbandry of laying hens with a greater proportion managed free-range in response to welfare concerns about cages [8].

*Campylobacter jejuni* (*C. jejuni*) and *Campylobacter coli* (*C. coli*) commonly colonise the gastrointestinal tract of poultry and are found in the faeces [13]. They are a major source of human food poisoning in Australia, the United States and Europe [14]. Studies of *Campylobacter hepaticus* have shown that it is also able to colonise the GIT and is present in the faeces of poultry [15, 16].

Confirmation of SLD by laboratory testing is important to differentiate it from other diseases such as fowl cholera and erysipelas which can have a similar history and gross pathology. Testing may also be used for SLD surveillance which could contribute to disease prevention and control. While culture of *C. hepaticus* from samples collected in the field is difficult [9] PCR testing has proved to be more sensitive and is able to detect *C. hepaticus* in bile samples, intestine content samples and faeces [15, 16].

PCR testing of clinical samples can be followed by high resolution melting (HRM) analysis where differences in curve shape and melting temperature correlate with variations in the sequence of amplicons. A major advantage of HRM analysis is that the variations in sequence can be detected without the need to sequence the amplicon [17–22]. The reagents required are relatively low cost and the technique is simple to perform [20].

The aim of this study was to develop and optimise a PCR–HRM assay suitable for clinical samples which was able to detect *C. hepaticus* and differentiate it from other poultry *Campylobacter* sp. in particular *C. jejuni* and *C. coli*.

## Materials and methods

### Ethics statement and sample collection

Approval for the sample collection from chicken was granted by the Charles Sturt University Animal Care and Ethics Committee (Protocol number A19046) and all experiments were performed in accordance with the relevant guidelines and regulations. Opportunistic chicken bile samples were collected from the gall bladder of 40 free-range egg layers found dead on farms or culled. Swabs of fresh faecal samples from 101 waterfowl including 66 Pacific black duck (*Anas superciliosa*) and 35 Australian wood duck (*Chenonetta jubata*) were collected from Wagga Wagg lake and lagoons and tested.

### *Campylobacter* strains

Two *C. coli* and three *C. jejuni* isolates, including three reference strains, ATCC33559, ATCC29428 and NCT11351 were used as the controls (Table 1). The *C. coli* (C669) and *C. jejuni* (BAL172236) isolates were isolated from chicken excreta or chicken carcasses and were provided by the Birling Avian Laboratories, New South Wales, Australia. All the isolates were cultured on non-selective sheep blood agar and incubated at 42°C under microaerobic

**Table 1. List of samples, origins, mean curve peak melting points ± SD, submitted GenBank accession numbers and genotypes of samples tested using PCR- HRM curve analysis.**

| Sample ID | Sample Type | Origin | Culture/species | Number of times tested | Melting point (˚C) | GCP ±SD [*] | Genotype [*] | Genotyped by sequence /GenBank accession number |
|---|---|---|---|---|---|---|---|---|
| B1 | chicken bile | Free range layer | Culture / *C. hepaticus* | 6 | 80.2±0.2 | 85.5±2.4 | *C. hepaticus* | *C. hepaticus*/MT682360 |
| B2 | chicken bile | Free range layer | Culture / *C. hepaticus* | 6 | 80.0±0.1 | 91.4±2.6 | *C. hepaticus* | |
| B3 | chicken bile | Free range layer | Culture / *C. hepaticus* | 6 | 80.3±0.0 | 93.1±6.1 | *C. hepaticus* | |
| B5 | chicken bile | Free range layer | Culture / *C. hepaticus* | 6 | 80.1±0.4 | 89.7±3.4 | *C. hepaticus* | |
| B6 | chicken bile | Free range layer | Culture / *C. hepaticus* | 6 | 80.0±0.0 | 79.2±8.2 | *C. hepaticus* | |
| B7 | chicken bile | Free range layer | Culture / *C. hepaticus* | 6 | 80.0±0.2 | 87.9±5.6 | *C. hepaticus* | |
| C1 | Chicken faeces | Free range layer | No culture | 10 | 80.0±0.0 | 90.3±4.8 | *C. hepaticus* | *C. hepaticus*/MT682361 |
| C2 | Chicken faeces | Free range layer | No culture | 10 | 80.2±0.0 | 91.6±2.1 | *C. hepaticus* | |
| C3 | Chicken faeces | Free range layer | No culture | 10 | 80.1±0.1 | 96.1±1.5 | *C. hepaticus* | |
| C4 | Chicken faeces | Free range layer | No culture | 10 | 80.0±0.2 | 88.9±4.4 | *C. hepaticus* | |
| C5 | Chicken faeces | Free range layer | No culture | 10 | 80.2±0.0 | 93.4±6.3 | *C. hepaticus* | |
| F2 | chicken bile | Free range layer | No culture | 8 | 80.0±0.3 | 88.6±5.2 | *C. hepaticus* | *C. hepaticus*/MT682362 |
| F8-1 | chicken bile | Free range layer | No culture | 8 | 80.1±0.1 | 87.2±5.5 | *C. hepaticus* | *C. hepaticus*/MT682364 |
| F8-2 | chicken bile | Free range layer | No culture | 8 | 80.1±0.3 | 90.2±2.1 | *C. hepaticus* | |
| F8-3 | chicken bile | Free range layer | No culture | 8 | 80.0±0.2 | 89.8±5.1 | *C. hepaticus* | |
| F9-1 | chicken bile | Free range layer | No culture | 8 | 80.2±0.1 | 90.1±1.4 | *C. hepaticus* | |
| F9-2 | chicken bile | Free range layer | No culture | 8 | 80.3±0.0 | 93.6±3.2 | *C. hepaticus* | |
| F9-3 | chicken bile | Free range layer | No culture | 8 | 80.0±0.0 | 90.4±4.8 | *C. hepaticus* | |
| F9-4 | chicken bile | Free range layer | No culture | 8 | 80.0±0.0 | 94.3±2.9 | *C. hepaticus* | |
| PR1 | chicken bile | Free range layer | No culture | 5 | 80.0±0.0 | 92.9±1.2 | *C. hepaticus* | *C. hepaticus*/MT682363 |
| PR2 | chicken bile | Free range layer | No culture | 5 | 80.0±0.0 | 87.5±2.8 | *C. hepaticus* | |
| PR3 | chicken bile | Free range layer | No culture | 5 | 80.2±0.0 | 84.3±2.1 | *C. hepaticus* | |
| PR4 | chicken bile | Free range layer | No culture | 5 | 80.3±0.1 | 72.5±1.1 | *C. hepaticus* | |
| PR5 | chicken bile | Free range layer | No culture | 5 | 80.4±0.1 | 87.4±2.8 | *C. hepaticus* | |
| PR6 | chicken bile | Free range layer | No culture | 5 | 80.4±0.1 | 82.4±4.1 | *C. hepaticus* | |
| PR7 | chicken bile | Free range layer | No culture | 5 | 80.4±0.1 | 89.2±2.4 | *C. hepaticus* | |

*(Continued)*

**Table 1.** (Continued)

| Sample ID | Sample Type | Origin | Culture/species | Number of times tested | Melting point (°C) | GCP ±SD [*] | Genotype [*] | Genotyped by sequence /GenBank accession number |
|---|---|---|---|---|---|---|---|---|
| PR8 | chicken bile | Free range layer | No culture | 5 | 80.3±0.2 | 78.5±5.1 | *C. hepaticus* | |
| PRC1 | chicken bile | Free range layer | No culture | 3 | 80.5±0.4 | 76.2±2.1 | *C. hepaticus* | *C. hepaticus*/MT682365 |
| PRC4 | chicken bile | Free range layer | No culture | 3 | 79.5±0.1 | 85.3±3.2 | *C. hepaticus* | |
| PRC5 | chicken bile | Free range layer | No culture | 3 | 80.3±0.2 | 73.7±7.2 | *C. hepaticus* | |
| PRC6 | chicken bile | Free range layer | No culture | 3 | 79.9±0.3 | 75.2±4.4 | *C. hepaticus* | |
| PRC8 | chicken bile | Free range layer | No culture | 3 | 80.1±0.2 | 77.5±8.3 | *C. hepaticus* | |
| PRC10 | chicken bile | Free range layer | No culture | 3 | 79.9±0.0 | 74.3±6.2 | *C. hepaticus* | |
| PRC11 | chicken bile | Free range layer | No culture | 3 | 80.3±0.1 | 72.0±0.2 | *C. hepaticus* | |
| PRC12 | chicken bile | Free range layer | No culture | 3 | 80.4±0.2 | 77.9±2.1 | *C. hepaticus* | |
| PRC13 | chicken bile | Free range layer | No culture | 3 | 80.5±0.1 | 93.0±5.2 | *C. hepaticus* | |
| UN1 | chicken bile | Free range layer | No culture | 3 | 80.4±0.1 | 90.3±2.7 | *C. hepaticus* | |
| UN2 | chicken bile | Free range layer | No culture | 3 | 80.4±0.1 | 85.6±3.8 | *C. hepaticus* | |
| 0283–1 | chicken bile | Free range layer | No culture | 3 | 80.3±0.4 | 74.5±4.8 | *C. hepaticus* | |
| 0283–2 | chicken bile | Free range layer | No culture | 3 | 80.2±0.2 | 76.3±5.4 | *C. hepaticus* | |
| PBD-13 | faeces | Pacific Black duck | No culture | 3 | 81.5±0.06 | 0.04±0.0 | *Variation* | *C. canadensis*/MW269513 |
| PBD-14 | faeces | Pacific Black duck | No culture | 3 | 81.6±0.04 | 0.00±0.0 | *Variation* | *C. canadensis*/MW269514 |
| ATCC29428 | Reference strain | Human faeces | Culture / *C. jejuni* | 12 | 81.7±0.0 | 0.19±0.3 | Variation | *C. jejuni* /MT682368 |
| ATCC33559 | Reference strain | Pig faeces | Culture / *C. coli* | 12 | 80.5±0.0 | 56.1±5.1 | Variation | *C. coli* /MT682367 |
| NCTC11351 | Reference strain | Bovine faeces | Culture / *C. jejuni* | 12 | 82.0±0.3 | 0.0±0.1 | Variation | |
| C669 | Chicken dropping | Broiler | Culture / *C. coli* | 12 | 80.7±0.1 | 51.7±5.4 | Variation | |
| BAL172236 | Chicken carcass | Broiler | Culture / *C. jejuni* | 12 | 81.8±0.1 | 0.3±0.1 | Variation | |
| HV10 | chicken bile | Free range layer | Culture / *C. hepaticus* | 16 | 80.3±0.1 | 98.3±1.1 | *C. hepaticus* | *C. hepaticus*/MT682366 |

[*] when *C. hepaticus* was used as reference genotype using a cut-off value of ≥72.

conditions (8% $N_2$, 4% $H_2$, 8% $O_2$ and 5% $CO_2$) for 72 hours. Pure cultures were then cultivated in broth medium for DNA extraction.

The field isolates were obtained from bile and faecal specimens collected at post mortem examination from the gall bladder and large intestine of chickens, respectively. The bile and

faecal specimens were collected into 5 mL sterile tubes which were stored in zip lock bags, on ice in a thermally insulated container until received at the laboratory where they were stored at -20˚C prior to DNA extraction. All isolates are listed in Table 1 with source details of *Campylobacter* species.

## DNA extraction

Total genomic DNA was extracted from bile specimens and bacterial cultures using Wizard® SV Genomic DNA purification system (Promega, Australia). The Quick-DNA Faecal/soil Microbe Miniprep Kit (Zymo Research, USA) was used to extract DNA from faecal samples, according to the manufacturer's instructions. Faecal swab samples of the same species of wild birds were pooled in groups of five before DNA extraction. All extracted DNA was quantified using the Nanodrop 2000 (Thermo Fisher Scientific, Australia). The concentration of each DNA sample was adjusted to 5 ng/μl for subsequent PCR or stored at -20˚C until use.

## PCR amplification

The *hsp60* gene was used to design the PCR primers (hsp60-F 5'- AAGAAATATTACAGC AGGAG-3' and hsp60-R 5'- GCATACCTTCAACAACATT-3'). The *hsp60* gene has been successfully used in PCR for detection of enteric bacteria and different species of *Campylobacter* [23–25]. The PCR amplification was performed in 25 μl reaction volume on a Rotor-Gene™ 6000 thermal cycler (Qiagen, Melbourne, Australia). PCR amplification was performed in 25 μl reaction volume and the reaction mixture contained 3 μl of extracted genomic DNA, 25 μM of each primer, 1.5 mM MgCl₂, 1250 μM of each dNTP, 5 μM SYTO 9 green fluorescent nucleic acid stain (Invitrogen, Australia), 5 x GoTaq Green Flexi Reaction Buffer and 1 U of Go Taq DNA Polymerase (Promega, Australia). The optimal PCR conditions were initial denaturation at 96˚C for 3 min, 35 cycles of 96˚C for 30 s, 55˚C for 30 s and 72˚C for 30 s, and a final extension of 72˚C for 5 min. In each set of PCR, HV10 *C. hepaticus* genomic DNA and distilled water were included as positive and negative controls, respectively.

## High-resolution melt curve analysis

On completion of PCR, HRM curve analysis was performed. The PCR products were subjected to 0.5˚C/s ramping between 50˚C and 90˚C. All samples were tested in triplicate and their melting profiles were analysed using Rotor-Gene 1.7.27 software and the HRM algorithm provided. Normalisation regions of 76–77˚C and 83.5–84.5˚C were applied for analysis.

The HV10 isolate (*C. hepaticus*) was set as a 'genotype' (GenBank accession CP031611), and the average HRM genotype confidence percentage (GCP) (the value attributed to each isolate compared with the genotype, with a value of 100% indicating an exact match) for the replicates was predicted by the software. The GCPs for *C. hepaticus* known isolates were averaged and the standard deviation (SD) calculated and used to establish the GCP range for the *C. hepaticus* cut-off point. The cut-off point was applied in HRM analysis to evaluate the differentiation power of the assay to discriminate the *Campylobacter* spp.

## Sequencing and nucleotide sequence analysis of PCR amplicons

Selected PCR amplicons (B1, C1, F2, PR1, PRC1, UN1, ATCC29428, ATCC33559 and HV10) were purified using the Wizard® SV Gel and PCR Clean-Up System (Promega, Australia) according to the manufacturer's instructions. Purified amplicons were sequenced in both directions with the same primers as used for PCR by Australian Genome Research Facility Ltd (AGRF Ltd., Brisbane, Australia). The sequences were analysed using Clustal W [26] and

DNASTAR (Meg Align) software. GenBank accession numbers were assigned to the nucleotide sequences of the *Campylobacter* isolates and reference strains (Table 1).

## Detection limit of the assay

The detection limit of the assay on DNA extracted from the bile sample was determined using dilutions of quantified *C. hepaticus* DNA. DNA extracted from *C. hepaticus* was serially diluted 10-fold from 1 ng to $10^{-8}$ ng. Each DNA dilution was tested in PCR. To evaluate the specificity of the assay, DNA was extracted from seven bacterial strains of genetically similar genera (*Klebsiella*, *Pseudomonas*, *Enterobacter*, *Staphylococcus*, *streptococci*, *E. coli* and *Pasteurella*) and tested in PCR-HRM.

## Results

Gel electrophoresis demonstrated that the PCR generated only amplicons of expected size of 269 bp and non-specific DNA amplification was not observed.

## Differentiation of *C. hepaticus*, *C. coli* and *C. jejuni* strains using conventional and normalised HRM curve analysis

The three *Campylobacter* species, (ATCC33559 *C. coli*, ATCC29428 *C. jejuni* and HV10 *C. hepaticus*) each produced one peak in conventional melt curve and distinctive normalised curves (Fig 1a and 1b). The mean melting points and SD of *C. coli*, *C. jejuni* and *C. hepaticus* were 80.50 ±0.00 ˚C, 81.65±0.00 ˚C and 80.25±0.14 ˚C, respectively.

## Non-subjective differentiation of *Campylobacter* species using GCP values and a mathematical calculation

The HRM curve analysis produced a distinct profile for *C. hepaticus* positive isolates. The positive control *C. hepaticus* (HV10) produced one peak and field samples (samples 1–40) also produced one peak at 80.15–80.40˚C. The *C. jejuni* samples (ATCC29428, NCTC11351 and BAL172236) produced a single peak at 81.75–82.0˚C, and the *C. coli* samples (ATCC33559 and C699) at 80.65–80.75˚C (Table 1). The negative control samples did not produce a peak. The conventional and normalised curves produced by *C. coli* samples was close to the curves range produced by the *C. hepaticus* isolates (Fig 2a and 2b).

Using GCPs of all *C. hepaticus* isolates, a cut-off value was generated as a mathematical model to assess the relationship of the isolates without requiring visual interpretation by the operator. The mean GCP of all *C. hepaticus* specimens was 92.4 and the mean SD was 4.1. The value of 5SD was subtracted from the average GCP to determine a cut-off point. A cut off point value of 71.9 was determined for *C. hepaticus* genotypes. Thus, the GCP range of the *C. hepaticus* isolates was determined to be 71.9–100 and was used for detection of all *Campylobacter* isolates. When *C. hepaticus* reference strain (VH10) was used as the reference genotype with a cut off value of 71.9, all *C. hepaticus* isolates produced a GCP (72–93) higher than 71.9 and genotyped as *C. hepaticus*. All *C. jejuni* and *C. coli* isolates also generated GCPs between 0.0–0.6 and 40.3–63.1, respectively, which were less than cut-off point (71.9) and therefore were automatically genotyped as 'variation'. When the *C. hepaticus* was used as reference genotype, the highest GCP values of *C. coli* and *C. jejuni* were 63.1 and 0.6, respectively. Therefore, the GCP gap between *C. hepaticus* and *C. coli* was about 9 and between *C. hepaticus* and *C. jejuni* was 71. To show the differentiation power of PCR-HRM in discriminating between *C. hepaticus* and *C. coli* and *C. jejuni*, the mean GCP values of each *Campylobacter* isolate was plotted in a dot plot which shows the GCP gap when *C. hepaticus* (HV10) is used as the

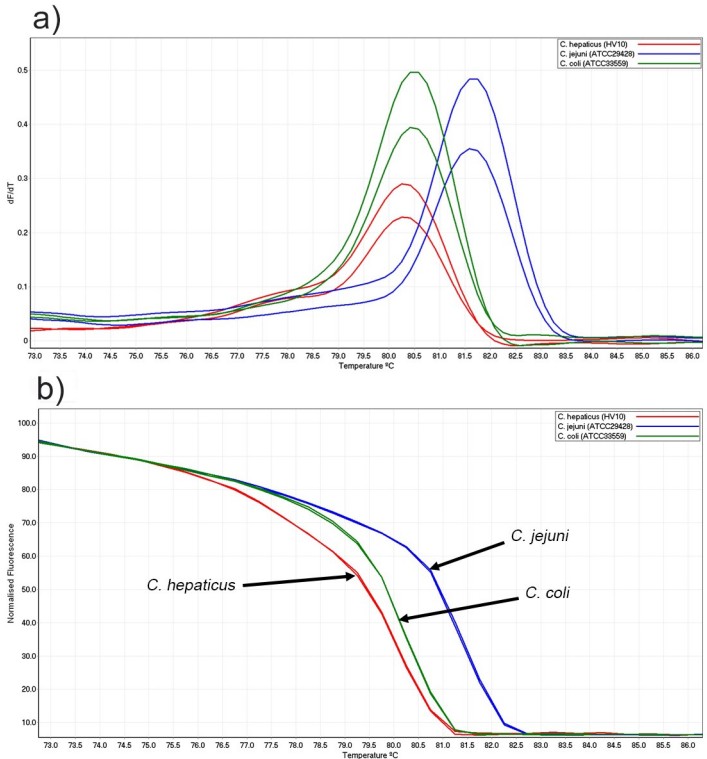

**Fig 1. Conventional and normalised melt curve analysis of three *Campylobacter* reference strains.** (a) Conventional and (b) normalised HRM curve analysis of PCR (*hsp60* gene) amplicons.

reference genotype (Fig 3). The PCR-HRM had a higher discrimination power in differentiating between *C. hepaticus* and *C. jejuni* isolates.

## Detection of minor variations in *hsp60* nucleotide sequence by the newly developed PCR-HRM curve analysis technique

To confirm that the differentiation of *Campylobacter* species by HRM curve analysis was related to variations in the nucleotide sequences of the amplicons, nucleotide sequences of amplicons from each distinct curve profile were determined and compared. The PCR amplicon for *C hepaticus*, *C. jejuni* and *C. coli* reference strains was 269 bp in length. One of *C. hepaticus* isolates (F8-1) had two substitutions (T to G and G to A) at positions 259 and 260, respectively. *C. coli* and *C. jejuni* strains each had 25 and 27 nucleotide substitutions when compared with *C. hepaticus* HV10 (Fig 4).

## Sensitivity and specificity of the assay

The concentration of DNA extracted from *C. hepaticus* positive control was measured using NanoDrop 2000 (ThermoFisher Scientific, Australia) and serial 10-fold dilutions were prepared. Each dilution was tested in PCR and 10 µl of each amplicon was run on 1.5% of agarose gel electrophoresis. The DNA sample was detectable up to $10^{-3}$ ng of *C. hepaticus* DNA (S1 Fig). None of seven bacterial strains (*Klebsiella*, *Pseudomonas*, *Enterobacter*, *Staphylococcus*, *streptococci*, *E. coli* and *Pasteurella*) produced a curve and remained negative using the PCR-HRM. The specificity of the assay was 100% as confirmed with DNA sequencing of *Campylobacter* species used in this study.

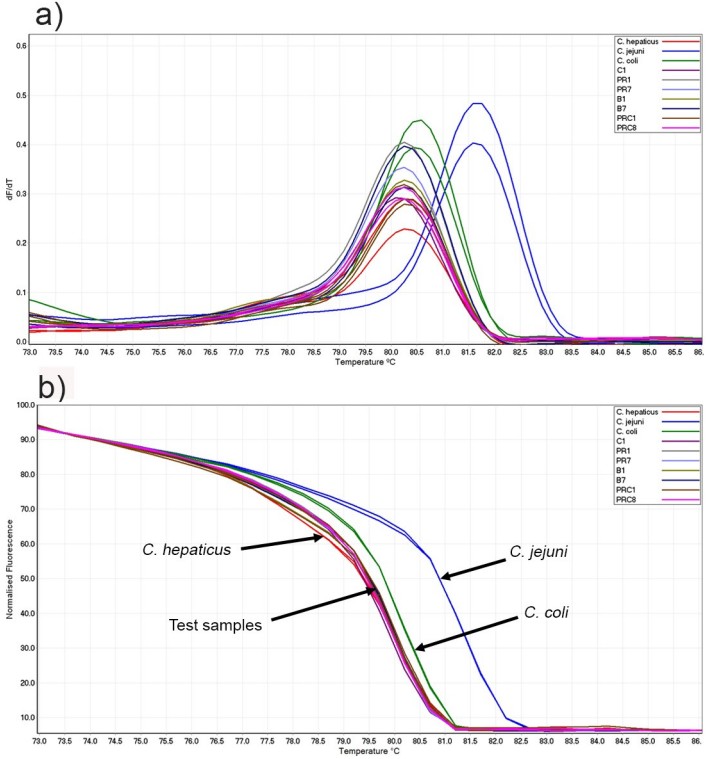

**Fig 2. Conventional and normalised melt curve analysis of *Campylobacter* strains and isolates.** a) Conventional and b) normalised melt curve analysis of PCR amplicons.

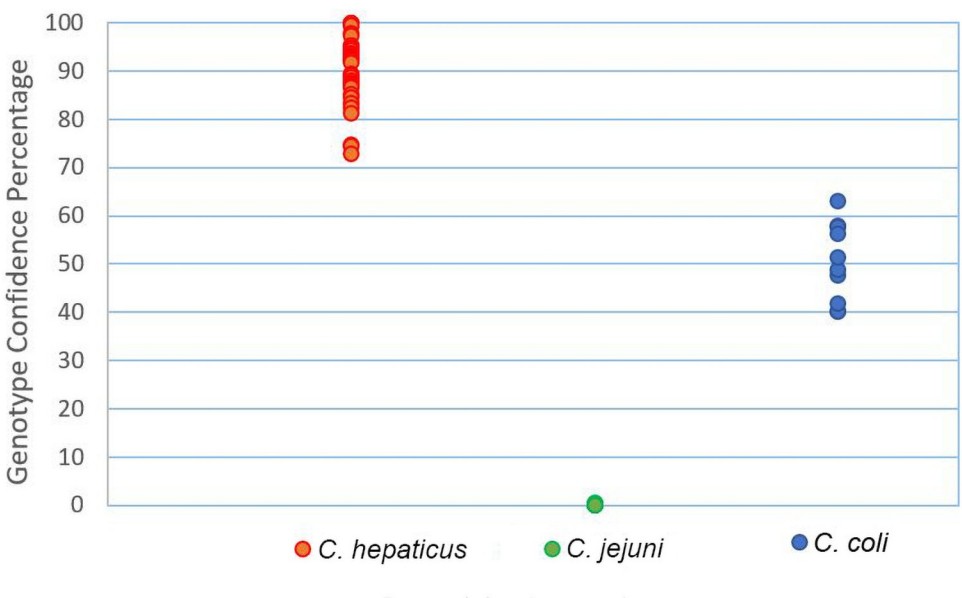

**Fig 3. Comparison of the distribution of GCPs from *C. jejuni* and *C. coli* isolates by individual value plot when *C. hepaticus* (VH10) was used as reference genotype.**

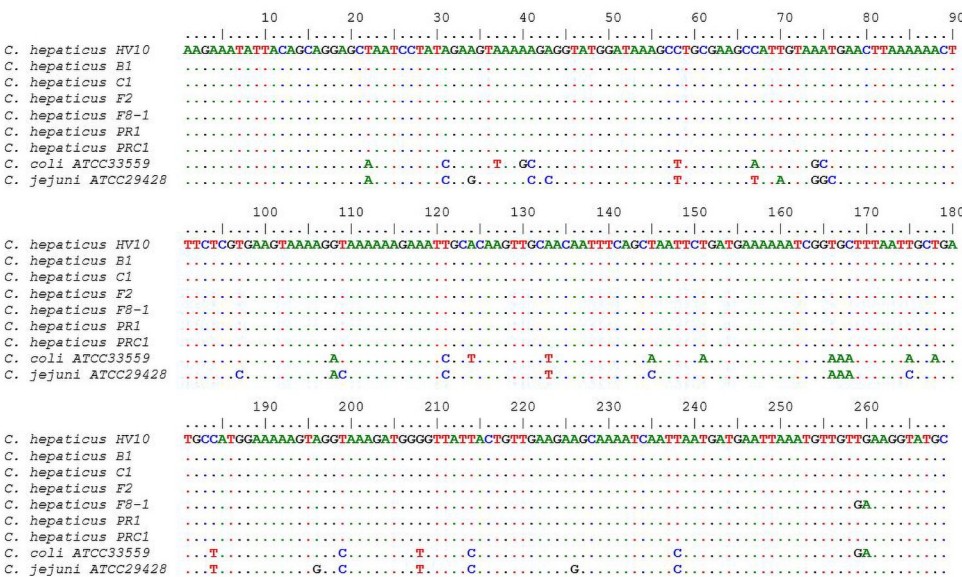

**Fig 4. Alignment of nucleotide sequences (*hsp60* gene) of selected *Campylobacter* isolates using CLUSTAL W.** Identical nucleotides and deletions are shown by '.' and '-', respectively.

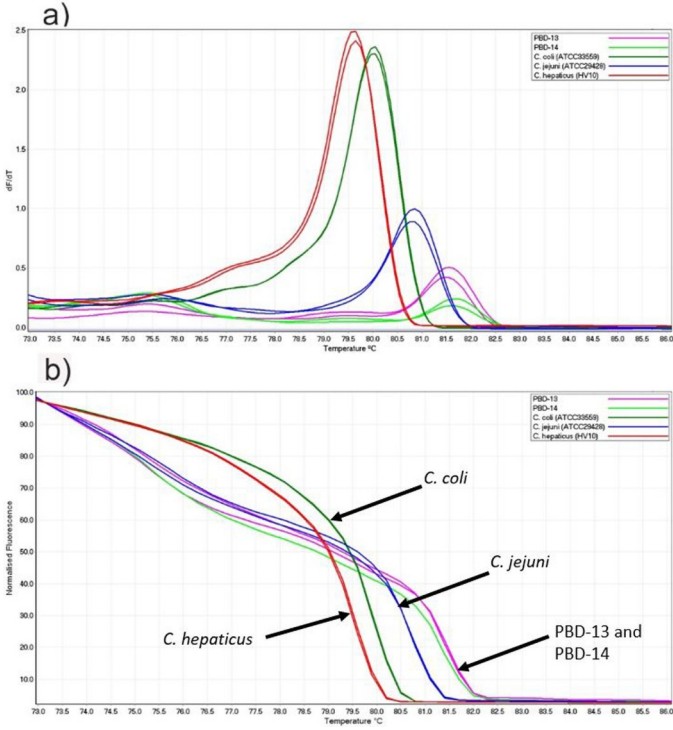

**Fig 5. Conventional and normalised melt curve analysis of *Campylobacter* species.** a) Conventional and b) normalised melt curve analysis of PCR amplicons from *Campylobacter* isolates detected in wild birds. All *campylobacter* spp. produced a single peak at different temperature. *Campylobacter* isolates from wild birds (PBD-13 and PBD-14) had melting points higher than *C. coli*, *C. jejuni* and *C. hepaticus*.

## Evaluation of PCR-HRM assay for differentiation of *Campylobacter* spp. in wild waterfowl

Out of 20 pooled samples (each pooled sample consisted of five faecal specimens of a particular wild duck species), two samples from Pacific black duck (PBD-13 and PBD-14) were found positive in PCR-HRM. Both samples produced similar melt curves (81.5±0.06) which were higher than melting points of *C. coli* (80.5±0.00), *C. jejuni* (81.2±0.02) and *C. hepaticus* (79.9 ±0.04) and therefore, generated distinct normalised curves (Fig 5). The amplicons of 272 bp were detected in the two groups of samples (both from Pacific black ducks) with identical nucleotide sequences (Fig 6). Sequence comparisons using BLAST showed 87.1% sequence similarity with *Campylobacter canadensis* (*C. canadensis*) and 85.8% similarity with *C. coli*. The phylogenetic analysis showed that *Campylobacter* species detected in wild waterfowls were genetically closer to *C. canadensis* when compared with *C. coli*, *C. jejuni* and *C. hepaticus* in their *hsp60* gene (Fig 6).

## Discussion

Spotty liver disease, caused by the bacteria *C. hepaticus*, is a disease of significance to the poultry industry, and may continue to increase in prevalence as the layer industry moves away from cages to free range and barn husbandry systems. Diagnosis of SLD can be difficult in the field as the majority of birds in a flock will appear to be in good health, with sick birds showing only mild depression or found dead [12]. To limit the spread of disease and to treat affected birds, the identification of the disease-causing pathogen needs to be rapid.

This study describes the development and assessment of a PCR with HRM curve analysis for the detection of *C. hepaticus*. Molecular tests are a more rapid method for detection of bacterial pathogens than traditional culture methods particularly for slow growing bacteria such as *C. hepaticus*. A PCR has been developed and optimised to detect *C. hepaticus* [16], and also a multiplex PCR has been developed which could simultaneously and specifically identify the presence of *C. jejuni*, *C. coli* and *C. hepaticus* [27]. While PCR and multiplex PCR are suitable for diagnosis of *C. hepaticus*, running samples on agarose gel electrophoresis is required in both methods to reveal the results which takes about one hour to conclude. While pre-poured agarose gels or buffer-free electrophoresis can save time, still loading samples and running the gel and obtain photographic record of the results is a time-consuming exercise. The use of HRM curve analysis is a relatively new technology which provides further automation by eliminating the need for gel electrophoresis and provides faster analytics to identify the genotype of samples in less than 15 minutes when compared to gel electrophoresis and saves time to pour, load and run the gel and therefore reduces time and labour-cost. The use of PCR-HRM curve analysis as a rapid genotyping method compared with PCR and gel electrophoresis has been well documented before [21, 28]. A multiplex-PCR and HRM curve analysis for differentiation of *C. jejuni* and *C. coli* suitable for use with poultry faecal or carcass swab specimens has also been reported [14].

This study assessed the use of bile specimens collected from dead or culled birds at post mortem examination for diagnostic purposes. *Campylobacter spp.* are recognised as bile resistant [29] and the level of *C. hepaticus* detected in the bile of hens has been shown to be considerably higher than in the liver [16], therefore bile was selected as the preferred sample for use in this study, however, a few faecal samples were also included in this study.

Targeting hsp60 gene in PCR for detection and differentiation of enteric bacterial pathogens including Campylobacter species has been reported to be a highly specific and reproducible approach [25, 30]. The specificity of this test was assessed by the inclusion of reference isolates of the two common *Campylobacter* spp, *C. coli* and *C. jejuni*, as well as *C. hepaticus*

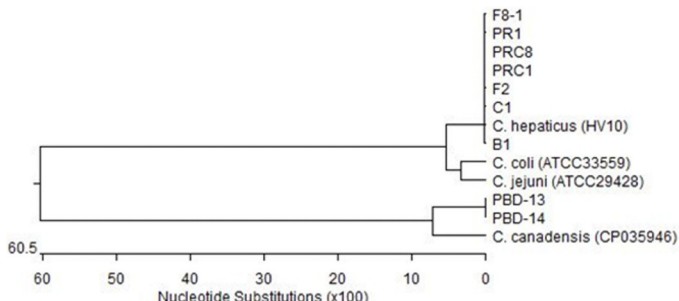

**Fig 6. Phylogenetic analysis of hsp60 gene sequences (269–272 bp) from *Campylobacter* species isolated from chickens (F8-1, PR1, PRC8, PRC1, F2, C1 and B1) and wild birds (PBD-13 and PBD-14).** Sequences were compared with those of *Campylobacter* reference strains. The tree was constructed using the Neighbour-joining method (DNASTAR software).

negative and positive field samples. This PCR with HRM curve analysis was able to correctly detect *C. hepaticus*, and differentiate *C. hepaticus* from *C. coli* and *C. jejuni*. The sensitivity of the test was determined using serial 10-fold dilutions of the *C. hepaticus* reference sample with the sensitivity determined to be $10^{-3}$ ng of *C. hepaticus* DNA. The sensitivity of PCR and culture in detection of *Campylobacter* species has been reported to be very similar [31].

The positive samples in PCR-HRM which were collected from wild Pacific black duck were genetically 87.1% similar to *C. canadensis* which has been reported in captive whooping cranes (*Grus Americana*) in Canada [32]. While partial *hsp60* gene sequences of wild bird samples (PBD-13 and PBD-14) show similarity with *C. canadensis*, full genome sequence of these samples is required for confirmation of the species. Among different *Campylobacter* species, *C. coli*, *C. jejuni*, *C. lari* and *C. hepaticus* are commonly associated with birds [8, 32–35]. While *C. coli*, *C. lari* and *C. jejuni* have also been reported in waterfowl, *C. jejuni* has been found more prevalent in faeces of waterfowl [36]. There is limited information on the biology of *C. canadensis* and its role in human health.

Banowary, Dang [14] concluded that PCR with HRM curve analysis is a powerful and valuable tool to study the nucleotide diversity of amplicons between tested specimens. The test enables detection of multiple species in the same test and is suitable for automation which makes this method a powerful diagnostic screening tool. There is also a reduction in the risk of cross-contamination as the PCR with HRM uses closed tubes, reducing the risk of DNA contamination of controls and samples [18].

This PCR-HRM curve analysis is suitable for the testing of birds suspected to have died due to SLD. This test would also be suitable for epidemiological studies at the processing plant, where a large volume of samples could be collected and tested to assess the prevalence of SLD on farms and regions. There is still considerable research required to understand the spread and pathogenicity of *C. hepaticus* and the test developed in this study could be useful to study disease prevalence. The PCR-HRM curve analysis showed both specificity and sensitivity for detection of *C. hepaticus*. This assay represents a rapid and sensitive diagnostic tool for assisting in the diagnosis of SLD in poultry.

## Supporting information

**S1 Fig. Sensitivity of the *C. hepaticus* PCR (10× serial dilutions) gel electrophoresis image.** Lane M: DNA ladder, Lane 1–9: *C. hepaticus* DNA at 1 ng, $10^{-1}$ ng, $10^{-2}$ ng, $10^{-3}$ ng, $10^{-4}$ ng, $10^{-5}$ ng, $10^{-6}$ ng, $10^{-7}$ ng and $10^{-8}$ ng concentrations.
(TIF)

## Acknowledgments

We thank Ms Lynette Matthews at Veterinary Diagnostic Laboratory (Charles Sturt University) for providing excellent technical assistance. We thank Professor Robert Moore and Dr Thi Thu Hao Van (RMIT University) for supplying *C. hepaticus* positive DNA and we also thank Ms Amy Crawshaw for helping in collecting samples from wild waterfowl.

## Author Contributions

**Conceptualization:** Tim Crawshaw, Seyed A. Ghorashi.

**Formal analysis:** Petrina Young, Pol Tarce, Sadhana Adhikary.

**Investigation:** Petrina Young, Pol Tarce, Sadhana Adhikary.

**Methodology:** Sadhana Adhikary, Joanne Connolly, Tim Crawshaw, Seyed A. Ghorashi.

**Resources:** Tim Crawshaw.

**Supervision:** Seyed A. Ghorashi.

**Validation:** Seyed A. Ghorashi.

**Writing – original draft:** Petrina Young, Seyed A. Ghorashi.

**Writing – review & editing:** Joanne Connolly, Tim Crawshaw, Seyed A. Ghorashi.

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
