## [Decision Letter · Decision Letter 0]

31 Mar 2021

PONE-D-21-00269

Evaluation of high-resolution melt curve analysis for rapid differentiation of Campylobacter hepaticus from other species in birds

PLOS ONE

Dear Dr. Ghorashi,

Thank you for submitting your manuscript to PLOS ONE. After careful consideration, we feel that it has merit but does not fully meet PLOS ONE’s publication criteria as it currently stands. Therefore, we invite you to submit a revised version of the manuscript that addresses the points raised during the review process.

The authors need to present evidence of specificity of the assay and time taken along with addressing other reviewer comments.

We look forward to receiving your revised manuscript.

Kind regards,

Iddya Karunasagar

Academic Editor

PLOS ONE

Additional Editor Comments:

The reviewers have pointed out number of issues that need to be addressed. The point regarding the specificity of the assay and time taken needs particular attention. Please revise considering all reviewer comments point by point.

Journal Requirements:

In your Methods section, please provide additional details regarding the chicken and wild birds, including the number and species, used in your study and ensure you have described the source. For more information regarding PLOS' policy on materials sharing and reporting, see https://journals.plos.org/plosone/s/materials-and-software-sharing#loc-sharing-materials.

In your Methods section, please provide additional information regarding the permits you obtained for the work. Please ensure you have included the full name of the authority that approved the field site access and, if no permits were required, a brief statement explaining why.

PLOS ONE now requires that authors provide the original uncropped and unadjusted images underlying all blot or gel results reported in a submission’s figures or Supporting Information files. This policy and the journal’s other requirements for blot/gel reporting and figure preparation are described in detail at https://journals.plos.org/plosone/s/figures#loc-blot-and-gel-reporting-requirements and https://journals.plos.org/plosone/s/figures#loc-preparing-figures-from-image-files. When you submit your revised manuscript, please ensure that your figures adhere fully to these guidelines and provide the original underlying images for all blot or gel data reported in your submission. See the following link for instructions on providing the original image data: https://journals.plos.org/plosone/s/figures#loc-original-images-for-blots-and-gels.

Your ethics statement should only appear in the Methods section of your manuscript. If your ethics statement is written in any section besides the Methods, please move it to the Methods section and delete it from any other section. Please ensure that your ethics statement is included in your manuscript, as the ethics statement entered into the online submission form will not be published alongside your manuscript.

We note that you have included the phrase “data not shown” in your manuscript. Unfortunately, this does not meet our data sharing requirements. PLOS does not permit references to inaccessible data. We require that authors provide all relevant data within the paper, Supporting Information files, or in an acceptable, public repository. Please add a citation to support this phrase or upload the data that corresponds with these findings to a stable repository (such as Figshare or Dryad) and provide and URLs, DOIs, or accession numbers that may be used to access these data. Or, if the data are not a core part of the research being presented in your study, we ask that you remove the phrase that refers to these data.

Reviewers' comments:

Reviewer's Responses to Questions

**Comments to the Author**

1. Is the manuscript technically sound, and do the data support the conclusions?

Reviewer #1: Partly

2. Has the statistical analysis been performed appropriately and rigorously? 

Reviewer #1: N/A

3. Have the authors made all data underlying the findings in their manuscript fully available?

Reviewer #1: Yes

4. Is the manuscript presented in an intelligible fashion and written in standard English?

Reviewer #1: Yes

5. Review Comments to the Author

Reviewer #1: PONE-D-21-00269: Evaluation of high-resolution melt curve analysis for rapid differentiation of Campylobacter hepaticus from other species in birds

The study evaluated high-resolution melt curve analysis to differentiate Campylobacter hepaticus from Campylobacter coli and Campylobacter jejuni. However, the authors have not conducted any other method comparison study to call the overall method “rapid.” Therefore, this rapid differentiation can be changed or removed or else should add about facts on time duration and justify accordingly.

The study does not include any experiment to find out the specificity of the assay. Also, the inclusion of other Campylobacter species and different related genus is important to assess the possibility of false-positive results.

In conclusion, “….rapid and sensitive diagnostic tool….” This can be rephrased or possibly soften appropriately

Change the figure no. to maintain the order

Lines 204-205: Rephrase this sentence

6. PLOS authors have the option to publish the peer review history of their article (what does this mean?). If published, this will include your full peer review and any attached files.

Reviewer #1: No

---

## [Author Response · Author response to Decision Letter 0]

7 Apr 2021

Academic Editor

Dear Prof. Iddya Karunasagar,

Thank you for your correspondence on 1st April 2021, regarding our manuscript Re: PONE-D-21-00269 (Evaluation of high-resolution melt curve analysis for rapid differentiation of Campylobacter hepaticus from other species in birds).

We are delighted that the manuscript is received well by the reviewers and we thank them for their constructive comments. We have answered reviewers’ comments and carefully revised our manuscript based on the reviewers’ feedback. The revisions in the manuscript are highlighted in track changes. 

We have also provided detailed responses to reviewers’ comments and concerns in a point-by-point manner, with their comments in red font. We also provided gel image as Supporting Information File.

We appreciate your time and effort during the evaluation process of our manuscript, and we look forward to hearing your editorial decision.

Kind regards

Seyed Ali Ghorashi

 

Journal Requirements:

 The manuscript was re-formatted based on PLOS ONE’s style.

2. In your Methods section, please provide additional details regarding the chicken and wild birds, including the number and species, used in your study and ensure you have described the source. For more information regarding PLOS' policy on materials sharing and reporting, see https://journals.plos.org/plosone/s/materials-and-software-sharing#loc-sharing-materials.

Required information including the number of tested chickens, wild birds, species and location of collected samples were added to the text lines 87-90. 

 The institutional permit for sample collection was added in the text lines 84-86. 

Gel image was provided as supporting information (S1 Fig) and the rest of figures were re-numbered accordingly.

Done.

 Your ethics statement should only appear in the Methods section of your manuscript. If your ethics statement is written in any section besides the Methods, please move it to the Methods section and delete it from any other section. Please ensure that your ethics statement is included in your manuscript, as the ethics statement entered into the online submission form will not be published alongside your manuscript.

 Done.

 The related phrase was removed from the text (line 179-180).

5. Review Comments to the Author

Reviewer #1: PONE-D-21-00269: Evaluation of high-resolution melt curve analysis for rapid differentiation of Campylobacter hepaticus from other species in birds

The study evaluated high-resolution melt curve analysis to differentiate Campylobacter hepaticus from Campylobacter coli and Campylobacter jejuni. However, the authors have not conducted any other method comparison study to call the overall method “rapid.” Therefore, this rapid differentiation can be changed or removed or else should add about facts on time duration and justify accordingly.

To address this comment, additional text has been added to the discussion lines 245-249 and 253-256 and provided references that compared the time required to complete PCR-HRM with PCR gel electrophoresis (Thomsen et al., 2012 and Sarker et al., 2014) that confirms PCR-HRM is rapid when compared with PCR gel electrophoresis.

The study does not include any experiment to find out the specificity of the assay. Also, the inclusion of other Campylobacter species and different related genus is important to assess the possibility of false-positive results.

To evaluate the specificity of the assay, DNA was extracted from seven bacterial strains of genetically similar genera (Klebsiella, Pseudomonas, Enterobacter, Staphylococcus, streptococci, E. coli and Pasteurella) and tested in PCR-HRM to address this comment. This is now added to the text in lines 154-157 and 215-218 of new version of manuscript.

In conclusion, “….rapid and sensitive diagnostic tool….” This can be rephrased or possibly soften appropriately

This comment is addressed above (first comment).

Change the figure no. to maintain the order

One figure (gel image- Fig 4) was provided as supporting information file and the rest of figures were re-numbered accordingly to maintain the order in the text.

Lines 204-205: Rephrase this sentence

To address this comment, the sentence was revised in lines 211-212 of new version of manuscript.

---

## [Decision Letter · Decision Letter 1]

26 Apr 2021

Evaluation of high-resolution melt curve analysis for rapid differentiation of Campylobacter hepaticus from other species in birds

PONE-D-21-00269R1

Dear Dr. Ghorashi,

We’re pleased to inform you that your manuscript has been judged scientifically suitable for publication and will be formally accepted for publication once it meets all outstanding technical requirements.

Kind regards,

Iddya Karunasagar

Academic Editor

PLOS ONE

Additional Editor Comments (optional):

The authors have addressed the reviewer comments satisfactorily.

Reviewers' comments:

Reviewer's Responses to Questions

**Comments to the Author**

1. If the authors have adequately addressed your comments raised in a previous round of review and you feel that this manuscript is now acceptable for publication, you may indicate that here to bypass the “Comments to the Author” section, enter your conflict of interest statement in the “Confidential to Editor” section, and submit your "Accept" recommendation.

Reviewer #1: All comments have been addressed

2. Is the manuscript technically sound, and do the data support the conclusions?

Reviewer #1: Yes

3. Has the statistical analysis been performed appropriately and rigorously? 

Reviewer #1: N/A

4. Have the authors made all data underlying the findings in their manuscript fully available?

Reviewer #1: Yes

5. Is the manuscript presented in an intelligible fashion and written in standard English?

Reviewer #1: Yes

6. Review Comments to the Author

Reviewer #1: (No Response)

7. PLOS authors have the option to publish the peer review history of their article (what does this mean?). If published, this will include your full peer review and any attached files.

Reviewer #1: No

---

## [Editor Report · Acceptance letter]

3 May 2021

PONE-D-21-00269R1 

Evaluation of high-resolution melt curve analysis for rapid differentiation of *Campylobacter hepaticus* from other species in birds 

Dear Dr. Ghorashi:

I'm pleased to inform you that your manuscript has been deemed suitable for publication in PLOS ONE. Congratulations! Your manuscript is now with our production department. 

Kind regards, 

on behalf of

Dr. Iddya Karunasagar 

Academic Editor

PLOS ONE